# Gut Microbiota, NAFLD and COVID-19: A Possible Interaction

Carmine Finelli

Department of Internal Medicine, ASL Napoli 3 Sud, Via di Marconi, 66, Torre del Greco, 80035 Naples, Italy; carminefinelli74@yahoo.it

**Abstract:** COVID-19, which is caused by SARS-CoV-2, is a major public health concern around the world. The beta coronavirus family includes SARS-CoV2, which enters cells via the ACE2 receptor. Patients in Wuhan, China, who suffered from the first sickness had no symptoms concerning the digestive system. Only 2.6% developed diarrhea, and only 2% had chronic liver illness. As the situation becomes more complicated, more people are reporting gastric issues. The disorder is characterized by diarrhea, anorexia, nausea, vomiting, abdominal discomfort and gastrointestinal bleeding, with diarrhea being the most common symptom. Several theories have been proposed for the genesis of gastrointestinal issues. Virus-induced cytopathic effects via ACE2, immune-mediated inflammatory cytokine storm, gut–lung axis function and drug-related damage are among them, not only in cases of COVID-19, but also in gastrointestinal illnesses.

**Keywords:** gut microbiota; NAFLD; COVID-19; obesity; ACE2 receptor; immune-mediated inflammatory cytokine storm; gut–lung axis function





## 1. Introduction

COVID-19, which is caused by SARS-CoV-2, is a major public health concern around the world. The beta coronavirus family includes SARS-CoV2, which enters cells via the ACE2 receptor [1]. Patients in Wuhan, China, who suffered from the first sickness had no symptoms concerning the digestive system. Only 2.6% developed diarrhea, and only 2% had chronic liver illness [2]. As the situation becomes more complicated, more people are reporting gastric issues. The disorder is characterized by diarrhea, anorexia, nausea, vomiting, abdominal discomfort and gastrointestinal bleeding, with diarrhea being the most common symptom [3]. Several theories have been proposed for the genesis of gastrointestinal issues. Virus-induced cytopathic effects via ACE2, immune-mediated inflammatory cytokine storm, gut–lung axis function and drug-related damage are among them [3], not only in cases of COVID-19, but also in gastrointestinal illnesses.

The liver is the body's largest digestive gland and is responsible for biligenesis and cleaning. Liver damage is a prevalent occurrence in COVID-19 patients. Direct viral infection, pharmacological cytotoxicity and an inflammatory immunological response are the most common causes of liver injury, according to histology and blood testing [4]. Hypoxic hepatitis, hepatic congestion induced by positive end-expiratory pressure (PEEP) and gut barrier failure are all possibilities [5]. The discovery of ACE2-positive cells in liver tissues, which makes the liver a potential target for SARS-CoV-2 infection, is one of the most important findings [4].

In addition, there is a global obesity epidemic that has resulted in insulin resistance, diabetes and chronic liver disease (CLD), which has become a serious public health concern [6,7]. Chronic hepatitis B and C, alcohol-associated liver disease and nonalcoholic fatty liver disease (NAFLD) are the most common causes of CLD. CLD can develop into inflammation (nonalcoholic steatohepatitis (NASH)), fibrosis and end-stage illnesses such as cirrhosis and hepatocellular carcinoma (HCC).

NAFLD is a global health issue that is the major cause of chronic liver disease in North America and Europe, affecting one out of every four persons [8]. The increased

incidence of NAFLD is mirrored by the rising prevalence of major NAFLD risk factors such as obesity, T2DM, CVD and metabolic syndrome [9]. NAFLD is a disease that can range from simple steatosis to NASH and cirrhosis, with about 25% of NAFLD patients at risk of developing NASH [9]. Patients with NAFLD who are infected with SARS-CoV-2 have a higher probability of severe COVID-19 outcomes, which is understandable considering the risk factors overlap. When patients with NAFLD were infected with COVID-19 and admitted to the hospital, they had a higher chance of developing COVID-19 disease, a higher likelihood of abnormal liver function tests (LFTs) from admission to discharge and a longer viral shedding period [10].

Additionally, COVID-19 development has been associated with NAFLD, a higher BMI, age and underlying comorbidities such as T2DM [11], suggesting that patients with NAFLD may be at a higher risk of COVID-19 disease progression and consequences once infected. Obesity-related poor immune responses to COVID-19 infection may be connected to this increased vulnerability to severe infection [12]. Meta-inflammation, or obesity-related chronic inflammation, is marked by pro-inflammatory responses mediated by the NLR family pyrin domain-containing 3 (NLRP3) inflammasome/interleukin (IL)-1 axis [13]. Furthermore, adipose tissue secretes pro-inflammatory tumor necrosis factor-alpha and IL-1 directly, resulting in downstream oxidative stress, and these pro-inflammatory pathways drive the pathogenesis of obesity-related diseases including NAFLD [14]. It is also possible that the stage of NAFLD fibrosis matters. In addition, in a recent comparison of obese patients with NASH to obese patients with steatosis, it was discovered that advanced stages of NAFLD may predispose patients to COVID-19 infection, as indicated by the considerably higher liver mRNA expression of ACE2 and transmembrane protease serine 2, genes linked to SARS-CoV-2 viral entry in obese patients with NASH [15,16].

Hepatocytes, which make up the majority of liver cells, are a major source of proteins implicated in innate and adaptive immune responses [17]. The liver maintains immunological homeostasis by limiting the passage of microbial and food antigens from the gut to the rest of the body, as well as synthesizing soluble molecules required for efficient immune responses [17]. By lowering the hepatic synthesis of proteins involved in innate immunity and pathogen-associated molecular pattern recognition, liver damage can affect immune surveillance. Immune dysregulation is a feature of both CLD and cirrhosis [18]. CLD inhibits the liver's homeostatic involvement in the systemic immune response. When damaged liver cells' molecular patterns are detected by circulating immune cells, they cause systemic inflammation in the form of activated circulating immune cells and increased serum levels of pro-inflammatory cytokines (e.g., TNF and IL-6). Furthermore, liver-related immunological dysfunction can make individuals more susceptible to infection. COVID-19-related morbidity and mortality are more common in patients with CLD, especially those with decompensated cirrhosis, which is to be expected in this situation [19].

SARS-CoV-2 infection has the potential to damage liver health in apparently healthy people, in addition to the negative prognostic impact of CLD in COVID-19 patients. COVID-19 patients typically have abnormal liver function tests, such as aspartate transferase (AST) and alanine transferase (ALT), according to current clinical evidence [20]. However, the processes underpinning COVID-19's impact on liver function are unknown and may be complex, and whether SARS-CoV-2 may infect hepatocytes has yet to be established. Because acute liver injury or cholestasis can occur in severe cases of cytokine storm unrelated to COVID-19, it is possible that immunological dysregulation caused by SARS-CoV-2 could play a role in COVID-19-related liver pathology [21]. The purpose of this paper is to evaluate and explain the implications of COVID-19 in patients with NAFLD, as well as the dangers of severe COVID-19 in patients with NAFLD and interactions with the gut microbiota.

Moreover, it has been shown that the associations between prior liver diseases and COVID-19 will contribute to worse clinical outcomes and should be taken seriously during care.

## 2. NAFLD and COVID-19 Progression

NAFLD/NASH will undoubtedly overlap with COVID-19 due to the enormous global prevalence of NAFLD and NASH, which affect around 25% of the global population [22]. Obesity and other lifestyle-related metabolic problems are linked to NAFLD (e.g., type 2 diabetes). Obesity and diabetes have been related to a poor prognosis since the beginning of the COVID-19 epidemic [23].

As a result of the significant global incidence of NAFLD, a large percentage of the population is at risk of developing severe COVID-19. Because NAFLD prognosis is predicated on the severity of liver fibrosis, it has been hypothesized that patients with NAFLD who have higher noninvasive liver fibrosis scores have a higher risk of severe COVID-19 [24]. Indeed, regardless of metabolic comorbidities, the incidence of severe COVID-19 is significantly higher in individuals with NAFLD who have been diagnosed with hepatic steatosis based on computed tomography or with an intermediate or high fibrosis-4 (FIB-4) index [25,26]. Patients with high or intermediate FIB-4 scores seem more likely to be older and obese, to have diabetes, to have elevated liver enzymes and C-reactive protein and to have lower levels of lymphocytes, platelets, triglycerides and HDL cholesterol than patients with low FIB-4 scores or individuals without NAFLD [27]. Furthermore, the necessity for mechanical ventilation is associated with obesity, diabetes mellitus and FIB-4 in COVID-19 patients, and FIB-4 is also associated with increased 30-day mortality [27]. The link between NAFLD/NASH and a severe course of COVID-19 suggests that a genetic risk score for NAFLD, based on the weighted effect of risk variants in PNPLA3 (patatin-like phospholipase domain containing 3)–TM6SF2 (transmembrane 6 superfamily member 2)–MBOAT7 (membrane-bound O-acyltransferase domain-containing protein 7)–GCKR [28,29].

Obese people and people with NAFLD/NASH may have increased hepatic expression of the SARS-CoV-2 receptors ACE2 and TMPRSS2, which could explain why these patients have a worse clinical course [15,21]. According to one study, NAFLD/NASH had no effect on the expression of the ACE2 and TMPRSS2 genes in the liver [13]. However, several studies suggest that chronic and progressive mechanisms, such as changed host angiotensin converting enzyme 2 (ACE2) receptor expression, direct viral attack, disruption of cholangiocyte function, systemic inflammatory reaction, drug-induced liver injury, hepatic ischemic and hypoxic injury and NAFLD/NASH-related glucose and lipid metabolic disorders, may play a role in the relationship between NAFLD/NASH and COVID-19 infection [30]. NAFLD/NASH and severe COVID-19 were also linked to gut microbiota and incomplete fatty acid oxidation products, potentially by regulating the host immune response [31]. It has been proven that the presence of NAFLD/NASH predisposes individuals to severe COVID-19 morbidity and mortality in the context of COVID-19 [32]. Several trials are being conducted in this clinical research field because there are no approved treatments for NAFLD/NASH. Nevertheless, because of the COVID-19 pandemic, several of these experiments have been halted. Because of the large global prevalence of CLD, ongoing NAFLD/NASH investigations should be continued [8].

## 3. Immune-Mediated Inflammatory Cytokine Storm

At this time, the pathophysiology of COVID-19 is unknown. Cytokine storms and cellular immune responses are thought to play a role in the onset and progression of illness [25,33]. A cytokine disturbance is a pathogenic inflammatory response to environmental stimuli that is aberrant and dynamic. Cells infected with SARS-CoV-2 release a considerable amount of inflammatory mediators and chemokines, which promote neutrophil aggregation. While neutrophils are primarily antiviral, their secretions, cytokines and chemokines also increase immune cell buildup, leading to overreaction. As a result, COVID-19 sufferers' immune systems are aberrant. Neurophilia was found in 34.5 percent of 197 individuals, which is recognized to be a trigger for the development of ARDS and sepsis in COVID-19 patients [34]. COVID-19 could possibly play a crucial role in the development of secondary hemophagocytic lymph histiocytosis (SHLH), an underrecog-

nized hyperinflammatory condition that can induce catastrophic and fulminant multiorgan failure and hypercytokinemia [35] (Figure 1).

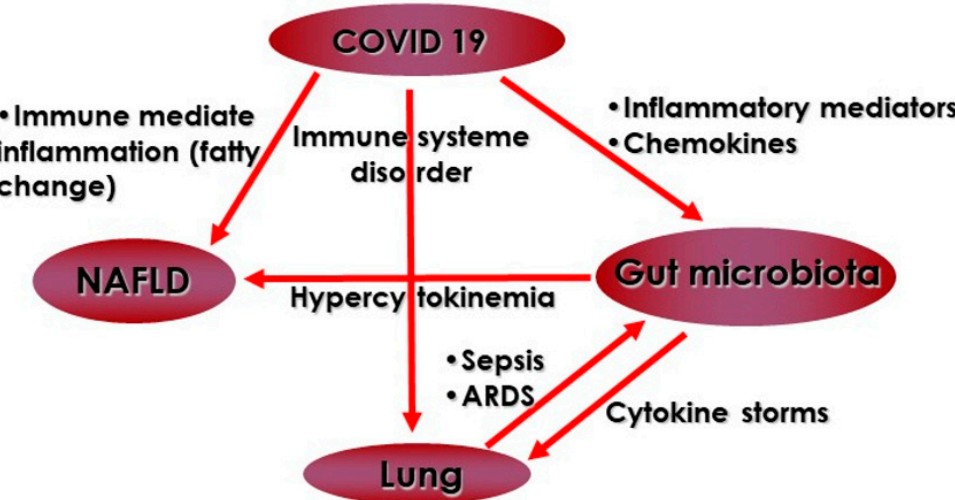

**Figure 1.** Possible interactions between gut microbiota, NAFLD and COVID-19.

Infected individuals had elevated levels of IL-1B, IFN-ᴛ, IP-10 and monocyte chemotactic protein 1 (MCP-1) expression, according to one clinical investigation [36]. These inflammatory cytokines can activate Type 1 helper cells and generate distinct immunological activation (Th1). Several studies have also found a link between cytokine levels in COVID-19 patients and disease severity [37]. In individuals with COVID-19, SARS-CoV-2 infection causes an inflammatory cytokine imbalance (Figure 1).

During COVID-19 progression, cytokine storm is linked to the development of ARDS and multiorgan failure outside of the lung [38]. COVID-19 patients, notably those with gastrointestinal disorders, may experience deterioration as a result of cytokine storms.

## 4. Pathogenic Links between the Gut Microbiota, NAFLD and Gut–Lung Axis in COVID-19

What role does the gut microbiota play in COVID-19 development? The gastrointestinal tract and the intestinal bacteria have a close relationship. The digestive and respiratory systems will impact each other through the gut–lung axis, notwithstanding their independence [39]. As a result, we believe that improving gastrointestinal microecology will lead to better COVID-19 patient predictions.

Even though a small case series from China indicated lower Lactobacillus and Bifidobacterium microbial dysbiosis in COVID-19 patients, no definitive research has established that the intestinal microbiota is connected to COVID-19 [40]. However, an earlier study revealed that the SARS-CoV-2 receptor ACE2 can regulate intestinal microbial homeostasis via amino acids [41]. Strong bacteria can ferment to produce short-chain fatty acids (SCFAs), which are mostly metabolized. SCFAs that have not been digested encourage the formation of naive CD4 + T cells, which help to control lymphocytes in the peripheral circulation and bone marrow. As a result, if the gut microbiota's stability is altered, the immunological response in the lungs will be compromised (Figure 1).

COVID-19 patients have higher levels of cytokines and inflammatory cells, which has been linked to sepsis and ARDS consequences [36,38]. Butyric acid, which is produced by the intestinal microbiota, has been demonstrated to reduce cytokine storms [42]. As a result, the gut microbiota may help to reduce the prevalence of sepsis and ARDS, both of which have a significant mortality risk in COVID-19. Furthermore, some researchers believe that the link between sepsis and gut microbiota abnormalities should be promoted together [43]. This suggests that gastrointestinal microbiota disorder causes sepsis, and that the stable structure of the intestinal microbiota is disrupted, resulting in the initiation

of a destructive cycle (Figure 1). Finally, we believe that the intestinal microbiota plays a role in avoiding and reducing COVID-19 problems.

SARS-CoV-2 is susceptible to binding particularly to ACE2 on hepatocytes, bile duct cells and liver endothelial cells to produce viral hepatitis damage, according to potential mechanisms of NAFLD in individuals with COVID-19 [44]. Aside from apoptotic liver cells, COVID-19 patients have greater fatty changes. NAFLD can be caused by immune-mediated inflammation and medication toxicity [45]. Severe COVID-19 could be more common in liver transplant recipients who take immunosuppressive medicines, particularly those who have metabolic problems (Figure 1).

## 5. Conclusions

In brief, the possible role of SARS-CoV-2 in the gastrointestinal system and liver should not be overlooked. It may enter cells directly through the ACE2 receptor, influencing the gastrointestinal system and the liver's normal function. Different pathways are also feasible, such as cytokine storm and the gut–lung axis. Meanwhile, digestive system diseases and COVID-19 are usually related, which might impair patient prognosis and raise the risk of death. The fundamental processes of COVID-19's association with digestive system illnesses are currently unknown. As a result, we anticipate that future research will concentrate on this topic and suggest more effective preventive measures, medicinal treatments and clinical procedures.

Because liver injury and CLD are linked to COVID-19 severity and mortality, indicators of liver disease, such as liver enzymes, liver fibrosis and liver steatosis, might be prioritized as COVID-19 severity prognostic markers. Individuals with liver illness have an immune inflammatory state that is especially important in other infectious disorders. This is especially important in light of the current global pandemics of NAFLD/NASH and COVID-19. Individuals with CLD (HCC, viral hepatitis, NAFLD/NASH and ALD) must also be screened and treated with special care. SARS-CoV-2 immunization should be prioritized for patients with CLD, particularly those with cirrhosis or extensive liver injury. Follow-up of vaccinated patients with CLD could provide further information about their compromised immune response. During the COVID-19 pandemic, services for patients infected with SARS-CoV-2 were restricted or postponed as a result of the SARS-CoV-2 pandemic. Such measures invariably have unintended consequences for patients with CLD.

These adverse reactions include delayed detection and treatment of a variety of liver disorders, which may increase CLD-related morbidity and mortality. Organ donation and liver transplants have decreased significantly as a result of the COVID-19 epidemic. COVID-19 is also having an impact on global efforts to eradicate viral hepatitis. Aside from the pandemic's detrimental impact on liver services, patient unhealthy behaviors have the potential to raise the global burden of liver disease in the near future. Additionally, on the other hand, it seems that SARS-CoV-2 infection causes only minor direct liver damage in those with healthy livers. However, the long-term effects of COVID-19 on the liver should not be overlooked, and more research is needed.

We propose that the gastrointestinal system and the liver are intimately linked to the formation and progression of COVID-19, particularly the putative interactions between gut microbiota, NAFLD and COVID-19, based on the above statement. However, because there are few studies currently accessible, this paper reviewed the pertinent viewpoints and provides plausible mechanisms. More research on COVID-19 patients is needed to gain better knowledge of the disease's pathophysiology and to find the best treatment for gut microbiota, NAFLD and COVID-19.

**Funding:** This research received no external funding.

**Institutional Review Board Statement:** Not applicable.

**Informed Consent Statement:** Not applicable.

**Data Availability Statement:** Not applicable.

**Conflicts of Interest:** The authors declare no conflict of interest.

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
