# Peer review of "Gut Microbiota, NAFLD and COVID-19: A Possible Interaction"

_2673-4168, doi:10.3390/obesities2020017_

Round 1

Reviewer 1 Report

Author provide a novel spot to link gut microbiota, NAFLD and COVID. But there are some minor point may need further demonstration:

  1. In line 124, author mentioned obese people and people with NAFLD/NASH may have increased hepatic expression of the SARS-CoV-2 receptors ACE2 that may explain some clinical observation, but also claim that NAFLD had no effect on the expression of ACE2. Could this be further clarified whether ACE2 is upregulated or not in these patients? If it's not ACE2 link COVID19 with NAFLD, then what is the possible mechanism?
  2. In Figure 1, according to the arrow, it's COVID19 cause NAFLD and change gut microbiota through immune related response, but it seems the other way described in the content. Should author adjust the figure or give more detail about how COVID19 worsen NAFLD and cause shift in gut microbiota?

Author Response

I thank you for your time to review my manuscript and providing valuable comments. I have considered each of the comments made and done the necessary changes to the best of my effort and understanding. A detailed response to each of the comments is presented below.

**********************************************************************************

Reviewer Comments to Author:

Reviewer: 1

Author provide a novel spot to link gut microbiota, NAFLD and COVID. But there are some minor point may need further demonstration:

  1. In line 124, author mentioned obese people and people with NAFLD/NASH may have increased hepatic expression of the SARS-CoV-2 receptors ACE2 that may explain some clinical observation, but also claim that NAFLD had no effect on the expression of ACE2. Could this be further clarified whether ACE2 is upregulated or not in these patients? If it's not ACE2 link COVID19 with NAFLD, then what is the possible mechanism?

However, several studies suggest that chronic and progressive mechanisms, such as changed host angiotensin converting enzyme 2 (ACE2) receptor expression, direct viral attack, disruption of cholangiocyte function, systemic inflammatory reaction, drug-induced liver injury, hepatic ischemic and hypoxic injury, and NAFLD/NASH-related glucose and lipid metabolic disorders, may play a role in the relationship between NAFLD/NASH and COVID-19 infection. [Xu, Yanlan et al. “Metabolic dysfunction associated fatty liver disease and coronavirus disease 2019: clinical relationship and current management.” Lipids in health and disease vol. 20,1 126. 3 Oct. 2021, doi:10.1186/s12944-021-01564-z.]     [30]

  1. In Figure 1, according to the arrow, it's COVID19 cause NAFLD and change gut microbiota through immune related response, but it seems the other way described in the content. Should author adjust the figure or give more detail about how COVID19 worsen NAFLD and cause shift in gut microbiota?                                                                                                      I apologize, I adjust the figure.

Reviewer 2 Report

Comments to the authors

  • L21-27: add appropriate references
  • L29-35: add appropriate references
  • L69-72: add appropriate references
  • L104-108: add appropriate references

Author Response

I thank you for your time to review my manuscript and providing valuable comments. I have considered each of the comments made and done the necessary changes to the best of my effort and understanding. A detailed response to each of the comments is presented below.

Comments to the Author:

  • L21-27: add appropriate references : Ozkurt Z, Çınar Tanrıverdi E. COVID-19: Gastrointestinal manifestations, liver injury and recommendations. World J Clin Cases. 2022;10(4):1140-1163. doi:10.12998/wjcc.v10.i4.1140    [3]
  • L29-35: add appropriate references: Kariyawasam JC, Jayarajah U, Abeysuriya V, Riza R, Seneviratne SL. Involvement of the Liver in COVID-19: A Systematic Review [published online ahead of print, 2022 Feb 24]. Am J Trop Med Hyg. 2022;106(4):1026-1041. doi:10.4269/ajtmh.21-1240     [4]
  • L69-72: add appropriate references: Neshat SY, Quiroz VM, Wang Y, Tamayo S, Doloff JC. Liver Disease: Induction, Progression, Immunological Mechanisms, and Therapeutic Interventions. Int J Mol Sci. 2021;22(13):6777. Published 2021 Jun 24. doi:10.3390/ijms22136777 [17]
  • L104-108: add appropriate references: Hegyi PJ, Váncsa S, Ocskay K, et al. Metabolic Associated Fatty Liver Disease Is Associated With an Increased Risk of Severe COVID-19: A Systematic Review With Meta-Analysis. Front Med (Lausanne). 2021;8:626425. Published 2021 Mar 12. doi:10.3389/fmed.2021.626425    [24]